# Noncovarying storage effect: Balancing and positive directional selection on mutant alleles that amplify random fitness and demographic fluctuations

Yuseob Kim [ID]*

Division of EcoScience, Ewha Womans University, Seoul, Korea

* yuseob@ewha.ac.kr

## Abstract

Temporally variable environments in natural populations generate fluctuations in both population size and the fitness effects of mutant alleles. The theory of storage effect, a species/allelic diversity-promoting mechanism discovered in ecology, predicts that rare mutants with fluctuating fitness can be positively selected and then maintained in balanced polymorphism if one part of the population, the 'field', is exposed to and the other, the 'refuge', is protected from fluctuating selection. A recent study found that oscillation in population size modifies the storage effect such that positive selection on a rare mutant occurs if its fitness and the size of the field change in the same directions. By this new version of storage effect, mutants with even smaller geometric mean fitness than the wild-type can be selected to intermediate frequencies and remain in balanced polymorphism or reach fixation. To further understand these eco-evolutionary dynamics and test their generality in natural populations, this study built more realistic models that assume randomly, not cyclically, fluctuating selection and common demographic features, including heterogeneous ecological patches or an age-structured population. Mathematical analysis elucidated that whether balanced polymorphism or fixation occurs depends on the relative magnitudes of demographic versus fitness fluctuations and that both results require the discordant oscillation in the population sizes of the field and refuge. Therefore this novel evolutionary force is called 'noncovarying storage effect (NSE)'. Multi-locus simulations revealed that oscillatory polymorphism can be maintained simultaneously or reach fixations at many loci. The latter occurs in a wider range of parameters and, if the fitness effects of mutations change the mean absolute fitness of the population, leads to positive feedback between demography and selection that causes a drastic amplification of population size fluctuation. These results suggest that the NSE is a potentially prevalent evolutionary force in nature for maintaining genetic variation or causing large demographic fluctuations.

**Data availability statement:** All relevant data for this study are within the paper, its Supporting information files, and publicly available from the GitHub repository (https://github.com/YuseobKimLab/RandomFluctuationEcoEvo).

**Funding:** This study was supported by the National Research Foundation (NRF) grant 2020R1A2C1009261 funded by the Korean government. The NRF had no role in study design, data collection and analysis, decision to publish, or preparation of the manuscript.

**Competing interests:** The author has declared that no competing interests exist.

## Introduction

Natural populations experience temporal changes in abiotic and biotic environmental factors. A randomly variable environment is expected to produce fluctuations in the fitness effects of non-neutral or phenotype-changing alleles segregating in the population. If the magnitude of fluctuation in relative fitness is large, evolutionary changes may occur rapidly at the pace of short-term ecological/demographic fluctuations. Recent developments in evolutionary biology have revealed that such rapid evolutionary changes are widespread [1–4]. Temporally variable environments should also generate fluctuations in various aspects of demography, most importantly the number of reproducing individuals (i.e., population size). One may then ask whether the mutual interaction between demographic and selective fluctuations creates qualitatively novel eco-evolutionary dynamics that cannot arise if one of the demographic or population genetic variables is held constant.

One of the major questions in the population genetics of fluctuating selection is whether different alleles that are favored at different time points can coexist in polymorphism. The prevalent conclusion has been that polymorphism is more difficult to maintain under temporal fluctuation in selection than without fluctuation, as the allele under selection is pushed closer to frequency 0 or 1, after which it is more readily lost or fixed in the population by genetic drift [5,6]. However, recent studies have suggested that temporally fluctuating selection can also act as balancing selection, an evolutionary force maintaining polymorphism at a level higher than expected under neutral evolution. It was shown that such balancing selection occurs if a population is partially protected from cyclical selection, that is if fitness fluctuation occurs only in a subset of the population [7–13]. Partial protection from fluctuating selection commonly occurs in many species. In spatially structured species, the amplitude of fluctuation in selective pressure is likely to be heterogeneous over the geographic range. In plants and animals that have their life cycles divided into several stages and reproduce in overlapping generations, selective pressure is unlikely to act uniformly on individuals of all stages or ages. For example, if a trait under fluctuating selection is expressed in the insect larval stage, alleles affecting this trait currently carried by adults are not visible to the selection. Similarly, alleles affecting traits that appear after germination remain neutral in the seed bank of a plant species. Such a subset of the population protected from fluctuating selection may be called a "refuge." The presence of various kinds of refuges was found to generate the advantage of rare alleles, an eco-evolutionary process commonly referred to as the storage effect [8,10,12–14]. A refuge acts to temporarily store the variants that otherwise would be removed from the population during unfavorable periods. More importantly, a refuge dampens the fluctuation in the mean absolute fitness (per-capita rate of reproduction) of individuals carrying the rare allele under selection, which increases the geometric mean fitness of the rare allele above that of the common allele [15]. The storage effect was first theorized in community ecology, shown to arise not only with partial protection from fluctuation provided by a refuge but with general covariance between environment and competition, and is now one of the most important ecological

theories for explaining the stable coexistence of species [7,16,17]. Imported into population genetics, it was recently proposed as one of a few mechanisms that may produce the genome-wide occurrence of oscillatory polymorphism, as observed in North American *Drosophila melanogaster* and other species, which appear to be maintained under fluctuating environments [18–20].

However, the previous study showed that, if the oscillation of population size is added to the model of the storage effect, a variant whose fitness also oscillates in the same directions can either be maintained in balanced polymorphism or be positively selected at all frequencies until it reaches fixation [15]. Such a variant may be called a fitness-amplifying allele because the relative fitness of an individual carrying this allele is high when the absolute fitness of all individuals is also high due to temporary population growth. Mathematical analysis showed that whether balancing or directional selection occurs on the fitness-amplifying allele depends on the relative strengths of fitness versus population size fluctuations. Previous studies on balancing selection by the storage effect in a constant-sized population required the coexisting variants to confer identical geometric mean fitness to the carriers. Such variants, exhibiting the geometric-mean-preserving fitness fluctuation (GMF) as defined in [15], are however selected either to loss or fixation in the presence of oscillation in the size of the population outside the refuge. One may therefore regard population size oscillation as a factor that destroys the potential of storage effect to generate balanced polymorphism. However, this population size oscillation led to balancing selection on a different type of mutants—alleles conferring a smaller geometric mean but identical arithmetic mean fitness compared to the wild-type. Since such AMF (arithmetic-mean-preserving fitness fluctuation) mutations, which are deleterious in a simple population (without a refuge), are probably more likely to occur in the genome than GMF mutations, this version of the storage effect modified by demographic fluctuation might be equally important or even more important in generating balanced polymorphism in nature. It was also shown that AMF variants are positively selected all the way to fixation if demographic fluctuation is relatively stronger than fitness fluctuation. Furthermore, it was found that, under a weak regulation of population density where the population size can over- or undershoot the carrying capacity, positive selection leads to the amplification of oscillation in mean absolute fitness, thus population size, due to the fixation of the fitness-amplifying allele [15]. In summary, the addition of population size fluctuation was found to generate novel eco-evolutionary dynamics not predicted by the conventional model of the storage effect.

The generality of novel eco-evolutionary dynamics discovered in [15] may, however, be questioned since the major result was obtained from the simulation of cyclically (seasonally) and symmetrically changing environments under which the population size and fitness oscillations are fully correlated. Although a mathematical analysis found general conditions for determining which evolutionary outcome (the loss, balanced polymorphism, or fixation of the fitness-amplifying allele) is expected, this result was based on the model of only two alternating seasons with fixed sizes. Therefore, further investigation using more realistic models is needed to obtain a better theoretical understanding and to determine whether the novel eco-evolutionary dynamics can arise in a substantially wider range of biological settings.

This study broadens the scope of investigation in the following directions. First, mathematical models assuming random, rather than deterministic and symmetrically seasonal, fluctuations in the environment were built and analyzed. A model with seasonal fluctuation is suitable for cases in which one cycle of environmental changes spans multiple generations. However, for many species, one generation is not shorter than a seasonal cycle. Random fluctuation of carrying capacities and the direction/strength of selection over generations, with a varying degree of correlation between these fluctuations, is probably relevant to a wider range of species including plants that exhibit mast seeding. Second, new models of population structures and inter-deme differences in selection intensity were investigated. Whereas [15] focused on a population with a seed bank [8,21] in which the mutant is completely protected from selection in the refuge, this study uses a model (the two-patch, or TP, model) that assumes two demes or ecological patches, both reproducing and undergoing changes in carrying capacity, under general heterogeneity in selection intensity. Selection in the refuge may not be completely neutral but simply weaker than that in the field. This TP model was also analyzed extensively by simulation to test its robustness against perturbations. Another model (the larval-subadult-adult, or LSA, model) assuming a population

with an age structure, in which the life cycle is divided into the larval (juvenile), subadult (young adult), and (old) adult stages, was also explored by simulation. Third, for both TP and LSA models, multi-locus stochastic simulations were performed. Simulations showed that mutant alleles causing wider fluctuation in offspring number can reach fixation at multiple loci across a wide range of conditions. These fixations were found to occur at faster rates with increasing numbers of loci, as fluctuating population size and fluctuating selection reinforced each other in positive feedback.

As will be clarified in the mathematical analysis, a key ingredient for this novel eco-evolutionary dynamic is the presence of a refuge whose size (or carrying capacity) remains relatively invariant or does not covary with the size of the rest of the population. One may therefore call this evolutionary mechanism the "noncovarying storage effect (NSE)." The result suggests that the NSE, arising under the general condition of heterogeneous fluctuating selection in structured populations, likely contributes to maintaining non-neutral genetic variation or to increasing demographic variability. Analytical and simulation results for the TP model will be presented first, followed by simulation results for the LSA model.

Although the majority of species to which the ecological scenarios of the TP and LSA models apply are diploids, haploid models will be constructed and analyzed for mathematical convenience. However, because the final outcomes of the eco-evolutionary dynamics are predicted by changes in allele frequency shortly after a mutant allele arises and when its frequency increased near fixation (i.e., whether a rare allele can invade the population), the results of this study using the haploid models should be relevant to diploid species as well. In a diploid species, homozygotes for the rare allele can be ignored. Consequently, the relative fitness of the rare allele in the haploid model corresponds to the fitness of heterozygotes relative to common homozygotes in the diploid population.

## Two-patch (TP) model

Consider a haploid population that reproduces in discrete generations. The outline of the life cycle is illustrated in Fig 1A. The population is divided into two subpopulations or patches that are subject to different levels and/or phases of fluctuating selection (thus the TP model). For convenience, these subpopulations will be called the field and the refuge, following

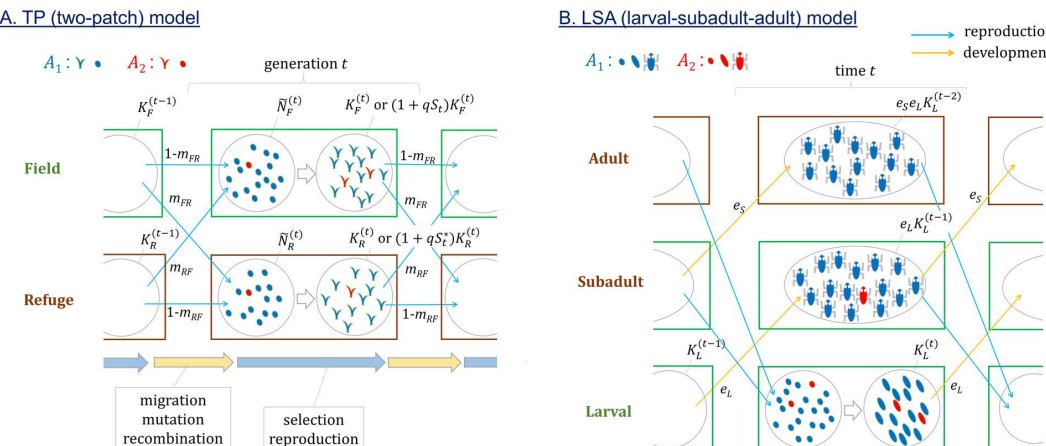

**Fig 1. Schematic diagrams of population models. A.** Two-patch (TP) model assumes a population of haploid individuals occupying a field, where the mutant allele $A_2$ has selective advantage $1 + S_t$ in generation $t$, and a refuge, where the advantage is $1 + S_t^*$. Each generation starts with young individuals that either migrated across or stayed in subpopulations after being reproduced in the previous generation. They also undergo mutation and recombination before selection starts. Then, they undergo selection such that the total number of reproducing individual is constrained to the carrying capacity $K_F^{(t)}$ for the field and $K_R^{(t)}$ for the refuge assuming soft selection. In case of hard selection the carrying capcities are effectively $(1 + qS_t)K_F^{(t)}$ and $(1 + qS_t^*)K_F^{(t)}$. **B.** In the larval-subadult-adult (LSA) model, at each time step haploid individuals move from larval to subadult stage with probability $e_L$ and from subadult to adult stage with probability $e_S$. Reproduction, followed immediately by mutation and recombination, occurs only in the subadult and adult stages. A new larval subpopulation at time $t$ starts with individuals produced at time $t$-1. At the end of time interval $t$, $K_L^{(t)}$ individuals remain in the larval stage before moving to the next stage.

the terminology in [15]. The latter refers to a subpopulation where selection on a mutant allele is weaker. It is assumed that the population is initially fixed for the ancestral $A_1$ allele at a locus. Then, a new mutant allele $A_2$ arises. The fitness of an individual carrying $A_2$ relative to $A_1$ is given by $1 + S_t$ in the field and $1 + S_t^*$ in the refuge, where $S_t$ and $S_t^*$ are drawn each generation from a joint distribution with $E[S_t] = E[S_t^*] = 0$. While the specific form of the distribution is needed for simulation (below), only its mean, variance, and covariance will be needed for deriving analytic solution in the next section. It should be noted that this study considers only AMF mutations; while the arithmetic mean of the fitness of $A_2$ over time is 1, its geometric mean is less than 1, in both subpopulations. Natural selection is thus expected to eliminate such an allele from a simple panmictic (unstructured) population.

A fluctuating environment causes fluctuations in the carrying capacities of both the field and refuge, which are given by $K_F^{(t)}$ and $K_R^{(t)}$ respectively for generation $t$. The carrying capacity is defined as the expected number of (reproductive) individuals at the end of each generation. It is assumed that the carrying capacities of both subpopulations depend on common environmental factors that fluctuate randomly over time, but that subpopulations may differ in the strengths of the dependence. The sizes of subpopulations, $N_F^{(t)}$ and $N_R^{(t)}$, after the reproduction step in generation $t$, may or may not match the corresponding carrying capacities depending on the strength of population density regulation (see below). Before a new generation starts, each individual in the field migrates to the refuge or remains in the field with probability $m_{FR}$ and $1 - m_{FR}$. Similarly, each individual in the refuge either migrates or not with probability $m_{RF}$ and $1 - m_{RF}$. Next, the size of the field (refuge) changes from $\widetilde{N}_F^{(t)}$ ($\widetilde{N}_R^{(t)}$) at the beginning of generation $t$ to $N_F^{(t)}$ ($N_R^{(t)}$) at the end, where $E[\widetilde{N}_F^{(t)}] = (1 - m_{FR}) N_F^{(t-1)} + m_{RF} N_R^{(t-1)}$ and $E[\widetilde{N}_R^{(t)}] = m_{FR} N_F^{(t-1)} + (1 - m_{RF}) N_R^{(t-1)}$.

Let $n_{1F}$ and $n_{2F}$ be the numbers of individuals in the field carrying $A_1$ and $A_2$ alleles at the beginning (before the reproduction step) of generation $t$ (thus $n_{1F} + n_{2F} = \widetilde{N}_F^{(t)}$). In the reproduction step, two different assumptions can be made regarding the effect of the mean fitness of individuals on the total population size. When the allele with higher or lower fitness increases in frequency, the total population size may not change due to an ecological constraint. In other words, the absolute fitness of the allele is frequency-dependent while its relative fitness remains constant. Conversely the mutant allele may increase or decrease the absolute reproductive output of the carriers regardless of its frequency such that the population size can increase over or decrease below the carrying capacity of a population in which all individuals carry the ancestral ($A_1$) allele. For convenience, these two modes of natural selection will be simply referred to as soft and hard selection, respectively, following the terminology in [22]. First, under soft selection, the final size of the field is constrained to $K_F^{(t)}$. The number of progeny born to each parent is given by $\frac{WK_F^{(t)}}{\overline{N}_F^{(t)}}$, where the relative fitness $W$ is 1 for $A_1$ and $1 + S_t$ for $A_2$, and $\overline{N}_F^{(t)} = n_{1F} + (1 + S_t) n_{2F}$. Second, under hard selection, fitness-changing mutations lead to changes in not only allele frequencies but also in the population size. To this effect, the number of progeny per parent is now $\frac{WK_F^{(t)}}{\widetilde{N}_F^{(t)}}$. Namely, when $q = n_{2F}/(n_{1F} + n_{2F})$ is the relative frequency of $A_2$ at the beginning of a generation, the size of the field is expected to reach $(1 + qS_t) K_F^{(t)}$, which becomes the effective carrying capacity of the field under hard selection. Similarly, reproduction in the refuge occurs either with soft or hard selection, where subscript $F$ in the above expressions is replaced by $R$ and $W$ is 1 for $A_1$ and $1 + S_t^*$ for $A_2$. The model so far described only the expected changes in the numbers of $A_1$- and $A_2$-carrying individuals. However, as reproduction in a finite-sized population was implicitly assumed, the role of genetic drift needs to be investigated. Stochastic simulation will be used for this investigation. In simulation, to set the variance of offspring number similar to that in the haploid version of Wright-Fisher model, each parent in both subpopulations is modeled to produce a Poisson number of offspring with the mean specified above.

Finally, how the environment fluctuates over time and how demographic and selective parameters—$K_F^{(t)}$, $K_R^{(t)}$, $S_t$, and $S_t^*$—respond need to be specified. The environmental condition is assumed to fluctuate randomly in each generation, therefore with no temporal autocorrelation. More specifically, fluctuations occur such that $K_F^{(t)} = K_{F0} e^{U_t}$ and $K_R^{(t)} = K_{R0} e^{V_t}$ where $U_t$ and $V_t$, together with $S_t$ and $S_t^*$, are drawn each generation from a joint normal distribution satisfying $E[U_t] = E[V_t] = E[S_t] = E[S_t^*] = 0$. The oscillation of carrying capacities is therefore symmetrical in the log scale, which is generally predicted in demographic models and is also consistent with empirical observation [23,24]. Various assumptions

can be made regarding the variance-covariance relationship among $S_t$, $S_t^*$, $U_t$, and $V_t$. However, the correlation among them satisfies that $Cov[S_t, U_t - V_t]$ is greater than $Cov[S_t^*, U_t - V_t]$, which gives the mathematical definition of the field and refuge. The specific joint distribution for these variables will be given below for simulation.

## Analytic results: the fate of the $A_2$ allele

This section aims to predict the fate of the rare $A_2$ allele—whether it will be lost, fixed, or increased to intermediate frequencies and maintained in polymorphism—as a function of the demographic and fitness parameters in the TP model. For convenience in the derivation, only the expected changes in the numbers of $A_1$- and $A_2$-carrying individuals will be tracked while ignoring genetic drift. In addition, a fixed scheme of migration, $m_{FR} = 1 - m_{RF} = r$, that sends a constant fraction $r$ of the population to the refuge regardless of the current location is assumed. (Below, the dynamics will be examined in simulation without this restriction in the migration rate.) First, soft selection, under which $N_F^{(t)} = K_F^{(t)}$ and $N_R^{(t)} = K_R^{(t)}$, is assumed. Let $n_1$ and $n_2$ be the total numbers of individuals (counted after reproduction/selection and before migration) carrying $A_1$ and $A_2$ in generation $t-1$ and $n'_1$ and $n'_2$ be their expected numbers in generation $t$. Then, with $n_1 \gg n_2$,

$$\frac{n'_2}{n_2} \approx (1-r)\frac{K_F^{(t)}}{\widetilde{N}_F^{(t)}}(1 + S_t) + r\frac{K_R^{(t)}}{\widetilde{N}_R^{(t)}}(1 + S_t^*) = \frac{(1 + S_t)K_F^{(t)} + (1 + S_t^*)K_R^{(t)}}{K_F^{(t-1)} + K_R^{(t-1)}} \tag{1}$$

(S1 Appendix). This equation takes into account that the change in the absolute number of $A_2$ is determined first by demography, as the size of field changes from $\widetilde{N}_F^{(t)}$ to $K_F^{(t)}$ and that of refuge from $\widetilde{N}_R^{(t)}$ to $K_R^{(t)}$, and also by fitness $1 + S_t$ and $1 + S_t^*$. Whether the copy number of rare allele $A_2$ is expected to increase or decrease is determined by the expectation of $\log\left[\frac{n'_2}{n_2}\right]$ over demographic and fitness fluctuations. It is shown in S1 Appendix that the condition for the $A_2$ allele to be positively selected (i.e., successfully invading the population initially fixed for $A_1$), $E\left[\log\left[\frac{n'_2}{n_2}\right]\right] > 0$, is simplified to

$$\frac{Cov[S - S^*, \ U - V]}{Var[S] + \rho^2 Var[S^*] + 2\rho Cov[S, \ S^*]} = \frac{Cov\left[S - S^*, \ log[\frac{K_F}{K_R}]\right]}{Var[S] + \rho^2 Var[S^*] + 2\rho Cov[S, \ S^*]} > \frac{1}{2\rho} \tag{2}$$

where $\rho = K_{R0}/K_{F0}$ (note that $K_F^{(t)} = K_{F0}e^{U_t}$ and $K_R^{(t)} = K_{R0}e^{V_t}$) represents the relative size of the refuge and super-/subscripts for time $t$ in variables were omitted. Note that, since $\rho > 0$, only positive covariance between $S - S^*$ and $U - V$ leads to positive selection on $A_2$.

If it is assumed that selection on $A_2$ in the refuge is weaker than that in the field by a ratio $\delta$, namely $S_t^* = \delta S_t$, the above inequality is simplified to

$$\Phi \equiv \frac{Cov\left[S, \ log[\frac{K_F}{K_R}]\right]}{Var[S]} > \frac{(1 + \delta\rho)^2}{2(1 - \delta)\rho} \tag{3}$$

Therefore, the rare $A_2$ allele is positively selected if the covariance between its fitness and the log-ratio of the field to refuge size is large relative to the amplitude of the fitness fluctuation. This requires that the demographic fluctuation in the population be sufficiently large, as already shown in [15], and that, to allow the substantial fluctuation of $\frac{K_F^{(t)}}{K_R^{(t)}}$, the carrying capacity of the refuge should not covary strongly with that of the field. Since this requirement for the refuge is essential and distinguishes the current from the previous models of the storage effect, the evolutionary force driving this positive selection may be called the noncovarying storage effect (NSE). Note that this term "noncovarying" refers to "not positively correlated", thus including the cases of negative covariance between $K_F^{(t)}$ and $K_R^{(t)}$, which lead to wide fluctuations in $\frac{K_F^{(t)}}{K_R^{(t)}}$ and thus large (positive) values of $\Phi$. In Discussion, the examples of noncovarying and covarying storages will be

presented. The conditions (2) and (3) are easier to meet as δ becomes small well below 1, namely, when the fitness fluctuation of $A_2$ becomes either smaller in the refuge compared to the field or negatively correlated to $S_t$.

In a similar manner, the condition for the $A_1$ allele to invade a population initially fixed for $A_2$ under soft selection becomes

$$\frac{Cov\left[S - S^*, \; UV\right]}{(1 + 2\rho)\,Var[S] + (2\rho + \rho^2)\,Var\left[S^*\right] - 2\rho Cov\left[S, \; S^*\right]} < \frac{1}{2\rho} \tag{4}$$

(S1 Appendix) or

$$\Phi < \frac{1 + \delta^2}{1 - \delta} + \frac{(1 + \delta\rho)^2}{2(1 - \delta)\rho} \tag{5}$$

if $S_t^* = \delta S_t$.

In a simpler case with δ = 0, from inequalities (3) and (5), it is predicted that the $A_1$ and $A_2$ alleles coexist in the population if $1/(2\rho) < \Phi < 1 + 1/(2\rho)$ under soft selection. With $\Phi < 1/(2\rho)$, the $A_2$ allele is eliminated from the population, as the fitness fluctuation that reduces its geometric mean in the field is more important than population size fluctuation. With $\Phi > 1 + 1/(2\rho)$, the $A_2$ allele is predicted to reach fixation in the population.

Using the same approach, the fate of the $A_2$ allele under hard selection can be analyzed. Note that the carrying capacities of the field and refuge are modified to $(1 + qS_t)K_F^{(t)}$ and $(1 + qS_t^*)K_R^{(t)}$ with hard selection where $q$ is the frequency of $A_2$. The frequency change of $A_2$ when rare is therefore effectively identical to that in soft selection. Therefore, the condition for $A_2$ invading the population fixed for $A_1$ is again given by inequalities (2) and (3). However, the condition for the fixation of $A_2$ is drastically different from that under soft selection. It is shown in S1 Appendix that the condition for $E\left[\log\left[\frac{n_1'}{n_1}\right]\right] > 0$ for rare $A_1$ is exactly the condition for $E\left[\log\left[\frac{n_2'}{n_2}\right]\right] < 0$ for rare $A_2$. Therefore, with $S_t^* = \delta S_t$, the condition for the $A_1$ allele to invade the population fixed for $A_2$ is approximately $\Phi < (1 + \delta\rho)^2/(2(1 - \delta)\rho)$. This means that the fixation of $A_2$ is ensured if $\Phi > (1 + \delta\rho)^2/(2(1 - \delta)\rho)$ once positively selected after its appearance by mutation, the $A_2$ allele is expected to increase in frequency all the way to fixation. This leads to the amplification of fluctuation in the size of the field, from $K_F^{(t)}$ to $(1 + S_t)K_F^{(t)}$. This result is consistent with the finding in Kim (2023) that, under weak density regulation of population size, an allele with a wider oscillation in absolute fitness reaches fixation in the presence of a refuge.

## One-locus simulation

To validate the above analysis and explore further eco-evolutionary dynamics in the TP model, a one-locus stochastic simulation was performed. The demographic and fitness fluctuations were modeled by $U_t = a_U Z_1 + b_U Z_2$, $V_t = a_V Z_1 + b_V Z_3$, $S_t = a_S Z_1 + b_S Z_4$, and $S_t^* = a_{S^*} Z_1 + b_{S^*} Z_5$ where $a_x$ ($b_x$) is a constant for determining the magnitude of change in variable $X$ due to a common environmental factor (other factors) and the $Z_i$s are standard normal variates that were drawn independently each generation. Therefore, $Z_1$ represents the state of the environment affecting both population sizes and fitness. To avoid negative fitness, the distributions of $S_t$ and $S_t^*$ were truncated at the lower bound of −0.999.

Simulations were first performed for a scenario in which the $A_2$ allele is completely protected from selection in the refuge ($S_t^* = 0$) and migration follows a simple scheme ($m_{FR} = m_{RF} = 0.5$). As suggested by the analysis above, the major parameter of eco-evolutionary dynamics in this case is $\Phi = \frac{Cov[S, \; U-V]}{Var[S]} = (a_S a_U - a_S a_V)/(a_S^2 + b_S^2)$. Simulation results were obtained for various values of Φ under soft (Fig 2A, B) and hard (Fig 2C, D) selection. Initially, it was assumed that the sizes of the field and refuge were finite and fluctuating but very large so that genetic drift can be ignored. Therefore the numbers of $A_1$ and $A_2$ individuals changed deterministically once the demographic and fitness parameters ($U$, $V$, and $S$) for a given generation were sampled from their joint distribution (Fig 2A). Various values of $a_S$ and $b_S$ were chosen so that

Φ ranged from 0.3 to 2, while ρ was set to 1. Starting from 0.5, the frequency of the $A_2$ allele at the 5,000th generation was recorded for 1000 replicates per parameter set. As predicted, with Φ ≤ 0.5, the frequency of $A_2$ approached 0, thus confirming negative selection, and with Φ ≥ 1.5 the frequency approached 1, thus confirming positive directional selection. The distribution of the allele frequency was approximately uniform between 0 and 1 when Φ = 1. Next, genetic drift and bidirectional mutation between two alleles were added to the above simulations (Fig 2B). This weakened the strength of all modes of selection. With Φ = 1, more simulation runs ended with allele frequencies close to 0 and 1. However, the proportion of runs ending with intermediate allele frequencies was still larger than that observed when the alleles were set as neutral (dashed curve in Fig 2B). Therefore, an evolutionary force to maintain polymorphism beyond the level under the neutrality, namely balancing selection, was confirmed.

Similarly, the fate of the $A_2$ allele under hard selection was examined. Stochastic simulation confirmed the approximation above predicting that $A_2$ would be positively selected towards fixation if $\Phi > \frac{1}{2\rho} = 0.5$, although genetic drift and mutation weakened the trend (Fig 2C, D). Positive selection on both rare and common $A_2$ under the same condition implies that this allele may not remain polymorphic for long in the population. Indeed, across all values of Φ there were fewer simulation runs ending with intermediate frequencies of $A_2$ compared to the simulation of neutral alleles subject to genetic

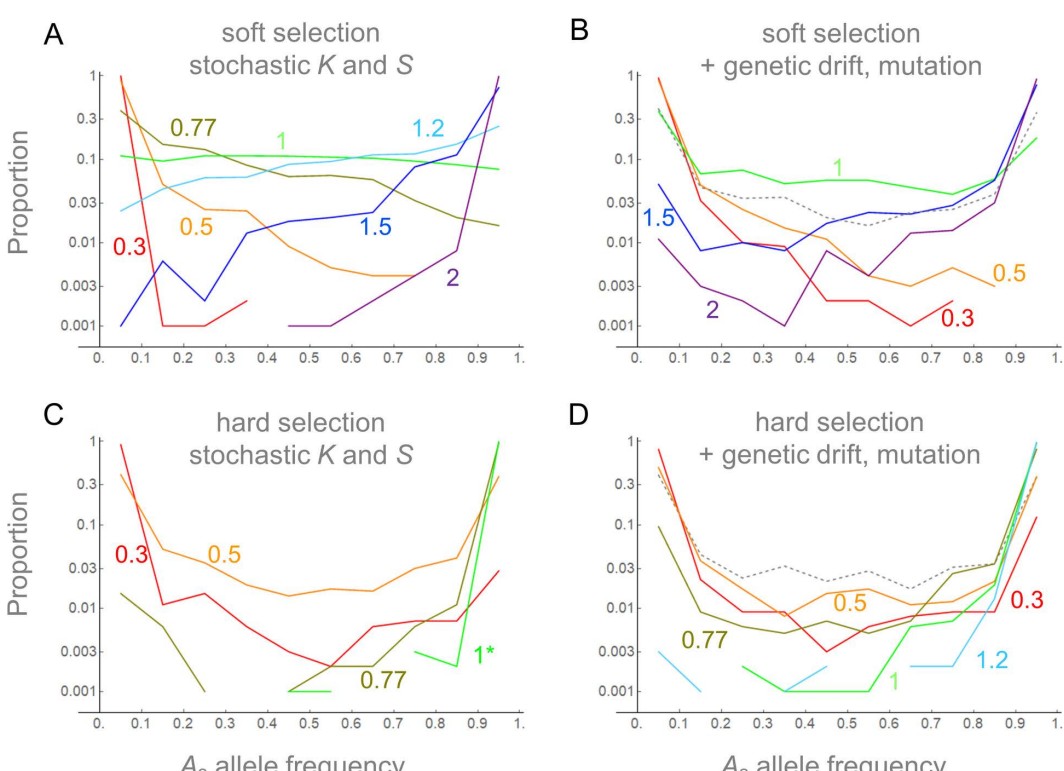

**Fig 2. Distribution of $A_2$ allele frequencies ($p$) at the 5,000th generation in the stochastic simulations of soft (A, B) and hard (C, D) selection in the TP model.** The initial frequency of $A_2$ is 0.5 in both field and refuge. Reproduction occurred while $K_F^{(t)}$, $K_F^{(t)}$ and $S_t$ varied stochastically without **(A, C)** or with **(B, D)** genetic drift and bidirectional mutation ($\mu = 5 \times 10^{-5}$). Proportions of simulation replicates (out of 1000) with the frequencies in the interval $0 \le p < 0.1$, $0.1 \le p < 0.2$, …, $0.8 \le p < 0.9$, and $0.9 \le p \le 1$ were plotted and connected by lines. Fluctuations of population sizes are given by $K_{R0} = K_{F0} = 1000$ ($\rho = 1$), $a_U = 0.3$, $b_U = 0.1$, $a_V = 0.05$, $b_V = 0.1$. Parameters of fluctuation were chosen to yield Φ = 0.3, 0.5, 0.77, 1, 1.2, 1.5, and 2: $(a_S, b_S)$ = (0.07, 0.23), (0.1, 0.2), (0.14, 0.16), (0.2, 0.1), (0.2, 0.04), (0.15, 0.05), and (0.1, 0.05). In the case of hard selection with Φ = 1 and no genetic drift (panel **C**), simulation ran only up to 2,500 generations: longer simulations or larger Φ resulted in all replicates with final frequencies > 0.9. The results connected by dashed lines in **B** and **D** represent the cases of neutral evolution ($a_S = b_S = 0$).

drift and bidirectional mutations only (dashed curve in Fig 2D). Therefore, balancing selection that was observed with soft selection did not arise with hard selection.

Next, the cases in which the $A_2$ allele is not completely neutral in the refuge were examined. A simple model setting $S_t^* = \delta S_t$ was used. This time, the $A_2$ allele is initially absent in the population but introduced with recurrent mutation. For a given parameter set, a simulation was run over at least $10^5$ generations. The mean $\overline{H}$ of expected heterozygosity ($2q(1-q)$, where $q$ is the relative frequency of $A_2$) and the proportion of time when $0 \leq q < 0.1$, $P_{01}$, and $0.9 < q \leq 1$, $P_{90}$, during simulation were recorded. Simulation with $S_t = S_t^* = 0$ but under the same demographic structure yielded $\overline{H}$ = 0.063 (average over $6 \times 10^5$ generations). Then, as balancing selection is defined to be a force yielding genetic variation above the neutral level, a result was classified as balanced polymorphism ("balanced" in Fig 3) if $\overline{H}$ > 0.075. If $\overline{H}$ < 0.05 and $P_{01} > P_{90}$, it suggests that negative selection against $A_2$ was the dominant evolutionary force. Such results were therefore classified as the loss of $A_2$ ("lost" in Fig 3), since recurrent losses of $A_2$ should be the major events in the simulation (confirmed by visual inspection of allele frequency trajectories; not shown). Conversely, results yielding $\overline{H}$ < 0.05 and $P_{90} > 0.7$, which indicates that positive directional selection on $A_2$ was the dominant force, were classified as the fixation of $A_2$ ("fixed" in Fig 3). All others were classified as not distinguishable from neutrality. Simulations largely confirmed the analytic approximations for the conditions for the loss, fixation, or balanced polymorphism of $A_2$ in a population initially fixed for $A_1$ (e.g., $\frac{(1+\delta\rho)^2}{2(1-\delta)\rho} < \Phi < \frac{1+\delta^2}{1-\delta} + \frac{(1+\delta\rho)^2}{2(1-\delta)\rho}$ for balanced polymorphism under soft selection and $\Phi > \frac{(1+\delta\rho)^2}{2(1-\delta)\rho}$ for the fixation under hard selection; Fig 3). It is shown that increasing $\delta$ above 0 makes it harder for balancing and positive directional selection on $A_2$ to occur; for a large value of $\delta$, the fluctuation (variance) of $S_t$ needs to be much smaller than its covariance with the fluctuation of $\log\left[\frac{K_F^{(t)}}{K_R^{(t)}}\right]$ (Eq 3). Small Var[$S_t$] then means that the relative strength of genetic drift affecting the evolutionary dynamics of $A_2$ is large, explaining the results not distinguishable from neutrality with large $\delta$ and $\Phi$ in Fig 3. Other results not distinguishable from neutral evolution were obtained when the predicted effects are near the boundary between balancing and directional selection. Another important result is that negative $\delta$, thus selection in the refuge being in the opposite direction, widens the parameter range for balancing selection.

Finally, the robustness of balancing and positive directional selection above to deviations from the simple migration scheme, $m_{FR} = 1 - m_{RF} = r$, was examined. In actual populations, migration rates may vary over time and may not be symmetrical between the field and refuge, at least on a short time scale. New simulations, therefore, sampled migration rates $m_{FR}$ and $m_{RF}$ from uniform distributions in ranges $[m_1, m_2]$ and $[m_3, m_4]$, respectively, each generation (Table A in S1 Table). With

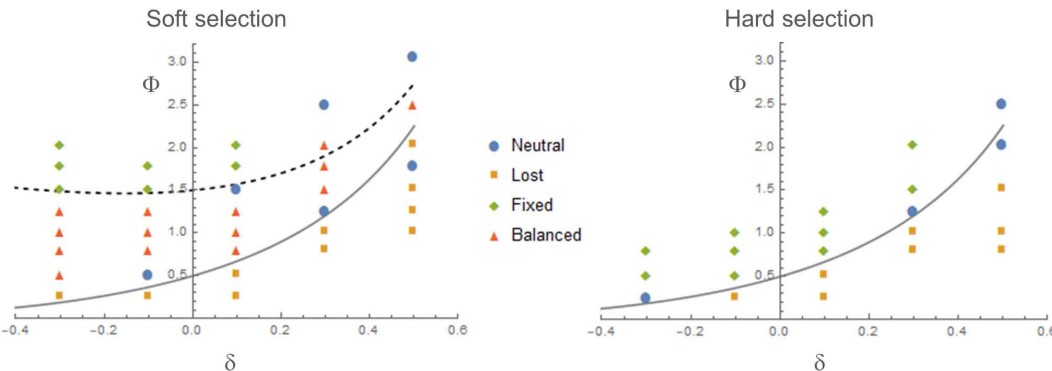

**Fig 3. Simulation results for non-zero values of δ.** For a given parameter set simulation started with the $A_1$ allele fixed in the population and ran for $10^5$ generations with recurrent bidirectional mutations and soft (left) or hard (right) selection. Four different outcomes regarding the overall behavior of the $A_2$ allele (balanced, lost, fixed, neutral) were marked by different symbols and plotted as functions of δ and Φ. The demographic parameters are identical to those in Fig 1. Five different values of δ (−0.3, −0.1, 0.1, 0.3, and 0.3) and ten different values of Φ were chosen: 0.246, 0.5, 0.8, 1, 1.25, 1.5, 1.786, 2.027, 2.5 and 2.94: ($a_S$, $b_S$) = (0.05, 0.22), (0.1, 0.2), (0.2, 0.15), (0.2, 0.1), (0.2, 0), (0.15, 0.05), (0.14, 0), (0.12, 0.02), (0.1, 0) and (0.085, 0). Gray and dashed curves plot $\Phi = \frac{(1+\delta\rho)^2}{2(1-\delta)\rho}$ and $\Phi = \frac{1+\delta^2}{1-\delta} + \frac{(1+\delta\rho)^2}{2(1-\delta)\rho}$ with $\rho = 1$.

$\rho = 1$ ($K_{F0} = K_{R0}$), limiting both $m_{FR}$ and $m_{RF}$ close to 0.5 yielded the highest level of balanced polymorphism under soft selection. However, allowing random deviation from 0.5 (e.g., $[m_1, m_2] = [m_3, m_4] = [0.3, 0.7]$) only slightly lowered the level of polymorphism. Smaller migration rates generally disrupted balancing selection. Interestingly, the effect of $m_{FR}$ and $m_{RF}$ was not symmetric; balancing selection remained effective with $m_{RF}$ close to 0.5 and $m_{FR}$ less than 0.2, however, not with the opposite ratio of rates (compare cases with $[[m_1, m_2], [m_3, m_4]] = [[0.1, 0.2], [0.4, 0.5]]$ and $[[0.4, 0.5], [0.1, 0.2]]$ in Table A in S1 Table). Similar effects of $m_{FR}$ and $m_{RF}$ on the directional increase of $A_2$ allele frequency under hard selection were observed.

In summary, one-locus simulations for the TP model not only confirmed the analytic approximations for the conditions under which the NSE leads to either balanced polymorphism or the fixation of the fluctuation-amplifying allele but also showed that the effect is robust to perturbations in the model such as genetic drift and randomness/asymmetry in the migration rate.

## Multi-locus simulation

To investigate how the eco-evolutionary dynamics generated by the NSE in the TP model unfold when there are many such loci in the genome, new sets of simulations were performed in which a haploid individual is modeled to have $L$ linked loci, each carrying either the $A_1$ or $A_2$ allele described above. To avoid exploring too many parameter dimensions while focusing on the effect of increasing the number of loci, $S_t^* = 0$ and a fixed migration scheme ($m_{FR} = m_{RF} = 0.5$) are assumed. The simulation tracks the numbers of individuals of different haplotypes (see S2 Appendix). Epistasis is not considered; the relative fitness of haplotype $i$ is given by $W_i = \prod_{l=1}^{L} (1 + S_{t,l})$, where $S_{t,l}$ is 0 if the $l$th locus of the haplotype carries $A_1$ and $a_S Z_1 + b_S Z_4$ if $A_2$. $Z_1$ is drawn once for all loci while $Z_4$ is drawn separately for different loci. Individuals randomly pair and undergo recombination by crossing-over such that the probability of recombination between adjacent loci is $c$ per generation. The simulation starts with a population fixed for $A_1$ at all loci. Then $A_2$ arises by mutations with probability μ/locus/generation.

Two major outcomes of the NSE – balancing selection and positive directional selection – were addressed. First, considering that (1) the negative frequency-dependent selection on the $A_2$ allele was caused primarily by its fitness being correlated to the fluctuation in the carrying capacity (population size) of the field and that (2) this demographic fluctuation should be "felt" at all loci in the genome, one may expect that such balancing selection can occur simultaneously at many loci harboring such variants. To verify whether this is true, simulations were run under soft selection and demographic/selective parameters that promoted polymorphism in the single-locus model (Φ = 1 for $\rho$ = 1) with $L$ = 1, 4, or 10. Exemplary trajectories of allele frequencies are shown in Fig A in S1 Fig. Each run lasted at least $10^5$ generations over which $\overline{H}$, the expected heterozygosity ($2p(1-p)$) averaged over time and loci, and the proportion of time the $A_2$ alleles spent in frequency between 0.1 and 0.5, $P_{15}$, and between 0.5 and 0.9, $P_{59}$, were recorded (Table 1). In all cases with fluctuating fitness ($a_S > 0$), $\overline{H}$ was above the level observed in equivalent simulations with neutral alleles ($S_t = 0$), thus confirming multi-locus balancing selection. For cases of stronger selection (larger $Var[S_t]$), the level of polymorphism per locus declined as $L$ increased from 1 to 10. The combined effects of the number of loci and the rate of recombination were not easy to interpret as they depended on the parameters of selection (for sets of fitness/demographic parameters that yield the same Φ = 1, $\overline{H}$ responded differently with varying $L$ and $c$). One possible explanation for generally lower variation with 10 loci instead of 1 or 4 is that the simultaneous segregation of alleles at many loci increased the variance of the offspring numbers, thus decreasing the effective population size and weakening the negative frequency-dependent selection that is needed for protected polymorphism. Further exploration of multi-locus polymorphism at a larger scale (in both number of loci and population size) may not be feasible with the current simulation approach based on haplotype frequencies.

Second, simulations were performed with parameter values under which directional selection for the $A_2$ allele is expected in the single-locus model (Φ > $1 + 1/(2\rho)$ for soft selection and Φ > $1/(2\rho)$ for hard selection). As expected, substitutions occurred sequentially until $A_2$ reached fixation at all $L$ loci (Fig 4A and Fig B in S1 Fig for example). Times (in generations) when the frequency of $A_2$ first exceeded 0.9 at each locus were recorded and sorted into $T_1$ to $T_L$. There

**Table 1. The long-term level of polymorphism in the multi-locus simulation.**

| Demography | Selection mode | $L$ | $c$ | $\overline{H}$ | $P_{15}$ | $P_{59}$ |
|---|---|---|---|---|---|---|
| $a_U=0.2$, $a_V=b_U=b_V=0.1$ | neutral ($a_S = b_S=0$) | 1 | – | 0.0639 | 0.069 | 0.074 |
| | soft selection, $\Phi=1$ ($a_S = 0.1$, $b_S=0$) | 1 | – | 0.132 | 0.15 | 0.16 |
| | | 4 | $10^{-4}$ | 0.0960 | 0.099 | 0.12 |
| | | 4 | $10^{-3}$ | 0.103 | 0.12 | 0.12 |
| | | 4 | 0.01 | 0.0977 | 0.11 | 0.12 |
| | | 4 | 0.1 | 0.136 | 0.11 | 0.14 |
| | | 10 | $10^{-4}$ | 0.080 | 0.087 | 0.095 |
| | | 10 | $10^{-3}$ | 0.102 | 0.12 | 0.12 |
| | | 10 | 0.01 | 0.109 | 0.15 | 0.10 |
| | | 10 | 0.1 | 0.107 | 0.13 | 0.12 |
| $a_U=0.3$, $a_V=0.05$, $b_U=b_V=0.1$ | neutral ($a_S = b_S=0$) | 1 | – | 0.0633 | 0.081 | 0.074 |
| | soft selection, $\Phi=1$ ($a_S = 0.2$, $b_S=0.1$) | 1 | – | 0.177 | 0.24 | 0.18 |
| | | 4 | $10^{-4}$ | 0.162 | 0.19 | 0.19 |
| | | 4 | $10^{-3}$ | 0.130 | 0.21 | 0.11 |
| | | 4 | 0.01 | 0.168 | 0.22 | 0.18 |
| | | 4 | 0.1 | 0.183 | 0.25 | 0.19 |
| | | 10 | $10^{-4}$ | 0.120 | 0.14 | 0.14 |
| | | 10 | $10^{-3}$ | 0.141 | 0.17 | 0.17 |
| | | 10 | 0.01 | 0.0934 | 0.076 | 0.14 |
| | | 10 | 0.1 | 0.105 | 0.093 | 0.15 |
| $a_U=0.05$, $a_V=b_U=b_V=0$ | neutral ($a_S = b_S=0$) | 1 | – | 0.0660 | 0.085 | 0.063 |
| | soft selection, $\Phi=1$ ($a_S = 0.05$, $b_S=0$) | 1 | – | 0.0816 | 0.11 | 0.081 |
| | | 10 | $10^{-4}$ | 0.0728 | 0.085 | 0.079 |
| | | 10 | $10^{-3}$ | 0.0982 | 0.12 | 0.11 |
| | | 10 | 0.01 | 0.0953 | 0.12 | 0.096 |
| | | 10 | 0.1 | 0.0952 | 0.13 | 0.092 |

Simulations were run for $5\times10^5$, $2\times10^5$, and $10^5$ generations with $L$ = 1, 4, and 10, respectively.

Other parameters: $K_{R0} = K_{F0}$ = 1,000 ($\rho$=1), $\mu$ = 2×10⁻⁵.

is a large difference in substitution dynamics between soft and hard selection. With soft selection, the mean waiting time, $\overline{T} = (T_1 + \ldots + T_L)/L$, increased moderately as $L$ increased from 1 to 8 (Fig 5A). This may be explained by a reduction in the efficacy of positive selection on $A_2$ at a given locus when selection at other loci results in genome-wide reduction in the effective population size.

With hard selection, however, $\overline{T}$ decreased as $L$ increased. Furthermore, the coefficient of variation, the standard deviation of $T_i$'s divided by $\overline{T}$, was significantly smaller with hard selection than with soft selection (Fig 5B). As expected for hard selection, the amplitude of the population size fluctuation progressively increased as the number of loci fixed for $A_2$ increased (Fig 4A). These results clearly demonstrate that positive feedback between demographic and selective fluctuations under hard selection led to a large amplification of population size fluctuation. Specifically, an initial fluctuation in $K_F^{(t)}$ positively selected the mutant allele $A_2$ at one locus whose fitness fluctuates in positive correlation with $K_F^{(t)}$, leading to the fixation of $A_2$ and therefore a larger fluctuation of the field (given by $(1 + S_t)K_F^{(t)}$), which makes it easier for the $A_2$ allele at another locus to be positively selected. This process could continue to increase the fluctuation of the field until its carrying

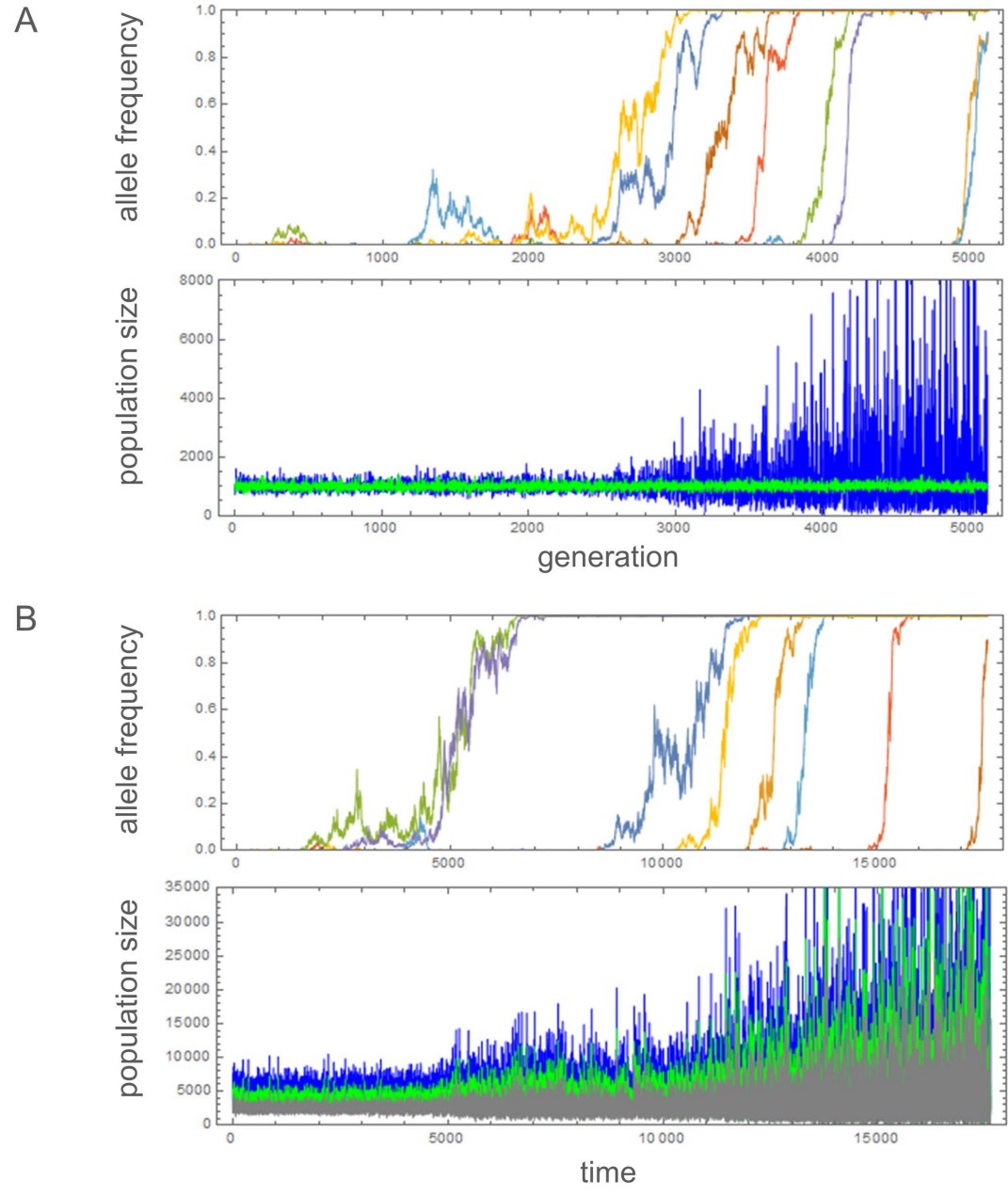

**Fig 4. Changes of mutant ($A_2$) allele frequencies and subpopulation sizes (A.** TP model, blue: $N_F^{(t)}$, green: $N_R^{(t)}$; **B.** LSA model, blue: larval, green: subadult, gray: adult population) in the multi-locus simulations of hard selection. Allele frequency trajectories of 8 loci were plotted in different colors. Parameters: **A** (TP model): $L = 8$, $c = 0.02$, $K_{R0} = K_{F0} = 1000$, $\mu = 2 \times 10^{-5}$, and $\Phi = 1$ ($a_S = 0.15$, $b_S = 0$, $a_U = 0.15$, $a_V = 0$, $b_U = 0$, $b_V = 0.1$). **B** (LSA model): $L = 8$, $c = 0.02$, $K_{L0} = 5000$, $\mu = 2 \times 10^{-5}$, and $\Phi' = 1$ ($a_S = 0.1$, $b_S = 0.1$, $a_L = 0.2$, $b_L = 0.1$).

capacity is given by $(1 + S_t)^L K_F^{(t)}$, where $L$ is the number of loci available in the genome for mutation into $A_2$. There was also a moderate effect of the recombination rate in accelerating the multi-locus fixation of $A_2$ with hard selection (Table B in S1 Table). Fixation times generally increased with decreasing recombination rates for both soft and hard selection, possibly because tighter linkage increased the occurrence of clonal interference.

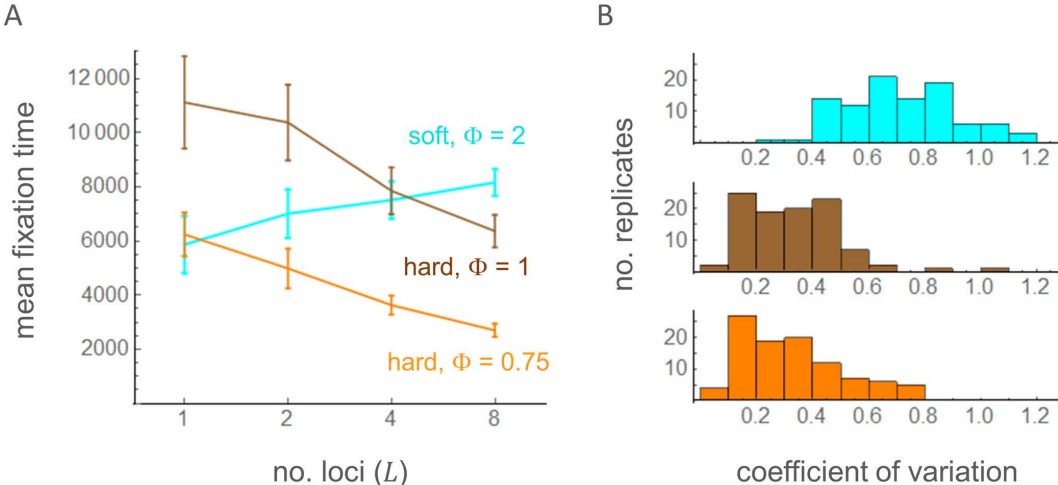

A

B

**Fig 5. Mean fixation time ($\overline{T}$) with varying number of loci (A; $L$ = 1, 2, 4, and 8) and the coefficient of variation in fixation times with $L$ = 8 in the multi-locus simulation of the TP model.** Error bars show two times the standard errors. A total of 100 replicates of simulations were run under soft selection with $\Phi$ = 2 (cyan; $a_S$ = 0.1, $b_S$ = 0, $a_U$ = 0.25, $b_U$ = $a_V$ = $b_V$ = 0.05), hard selection with $\Phi$ = 1 (brown; $a_S$ = 0.1, $b_S$ = 0, $a_U$ = 0.1, $a_V$ = 0, $b_U$ = $b_V$ = 0.1), or hard selection with $\Phi$ = 0.75 (orange; $a_S$ = 0.2, $b_S$ = 0, $a_U$ = 0.25, $a_V$ = 0.1, $b_U$ = $b_V$ = 0.05). Other parameters are identical to those in Fig 4A.

## Larval-subadult-adult (LSA) model

Although the refuge in the TP model is expected to play the role of various demographic/genetic elements that were previously shown to produce the storage effect, such as seed banks, diapausing stages, or a particular life stage in a population with overlapping generations, the noncovarying storage effect found in the TP model may not be replicated in other population structures as the nature of the refuge (i.e., whether it undergoes its own reproduction and whether the lineage of an allele can persist in a refuge for many generations) and the pattern of 'migration' in and out of the refuge can be qualitatively different. I therefore investigate an explicit discrete-time model of an age-structured population with overlapping generations. Three stages – 'larval', 'subadult', and 'adult' – in the life cycle of a haploid species are assumed. Subadult means the early stage of reproducing adults. Time is counted in intervals of the length required for one life stage to advance to the next. At the end of each interval, the final size of larval population is regulated to its carrying capacity ($K_L^{(t)}$ at time $t$). Probabilities for an individual to advance from larval to subadult and from subadult to adult stages are given by $e_L$ and $e_S$, respectively. Therefore, the sizes of the subadult and adult populations are expected to be $e_L K_L^{(t-1)}$ and $e_L e_S K_L^{(t-2)}$ at time $t$. $K_L^{(t)}$ is assumed to vary randomly over time with no autocorrelation. Specifically it is modeled to be $K_L^{(t)} = K_{L0} exp(a_L Z_1 + b_L Z_2)$, where $Z_1$ and $Z_2$ are standard normal variables drawn at each time step.

The population is made of haploids carrying either the $A_1$ or $A_2$ allele at each of $L$ linked loci. The (marginal) fitness of $A_2$ relative to $A_1$ at locus $j$ is given by $1 + S_{j,t}$ ($j$ = 1, ..., $L$), where $S_{j,t} = a_S Z_1 + b_S Z_{3,j}$. $Z_{3,j}$ is drawn independently for different loci at each interval $t$. Then, the fitness of haplotype $i$, $W_i^{(t)}$, is given by the multiplication of allelic fitness across all loci. It is assumed that this haplotype fitness determines the viability in the larval stage only; the carriers of two alleles do not differ either in the probability of advancing to the next stage or in fertility in the subadult and adult stages. If there are only larval and subadult stages, they constitute a population that reproduces in effectively discrete generations. The addition of the (old) adult stage provides a storage for alleles regardless of their reproductive success in the previous (subadult) stage. Therefore, the adult stage plays the role of the refuge and is an example of a noncovarying storage because its size is independent of the concurrent subadult stage.

Multi-locus simulations, allowing not only genetic drift in the reproduction step but also stochastic advances from one to the next life stage (see S2 Appendix for method), were performed with limited sets of parameters, each starting with a

population fixed for haplotype 0, which carries $A_1$ at all loci. The evolutionary dynamics of the $A_2$ alleles in each simulation run over 50,000 time units were summarized by the mean heterozygosity and the proportion of time $A_2$ remained close to relative frequency 0 or 1 (Table C in S1 Table). Major results produced in the TP model – multi-locus balancing selection under soft selection or positive directional selection on $A_2$ under hard selection – were readily obtained with this model. In agreement with [15] and the TP model, the relative strength of demographic versus fitness fluctuation determined the outcome; when we define $\Phi' = \frac{Cov\left[log[K_L^{(t)}/K_{L0}],S_{j,t}\right]}{Var[S_{j,t}]} = \frac{a_L a_S}{(a_S^2 + b_S^2)}$, which is analogous to $\Phi$ in the TP model, positive selection on rare $A_2$ occurred when $\Phi'$ was above 0.7. With soft selection and above this threshold of $\Phi'$, the frequency of $A_2$ underwent sustained oscillations that are indicative of balancing selection (Fig C in S1 Fig). Then, as $\Phi'$ continued to increase, balancing selection transitioned into positive directional selection. With hard selection, positive directional selection occurred once $\Phi'$ increased above the threshold. Exemplary replicates show that, as the $A_2$ alleles reach fixation across loci, fluctuations in the sizes of all three life stages become amplified (Fig 4B, Fig C in S1 Fig). Negative (positive) correlation between mean fixation times and the number of loci under hard (soft) selection and the clustering of fixation times with hard selection were also observed in other sets of simulations (Fig D in S1 Fig), which basically replicated the results from the TP model (Fig 5). This confirmed that positive feedback between demographic and fitness fluctuations under hard selection occurs in both models. In conclusion, all major aspects of the NSE found in the TP model were shown to arise in the LSA model as well.

## Discussion

A key assumption in this study is that a trade-off (antagonistic environmental pleiotropy [25]) commonly occurs for the fitness effects of a mutant allele between different time points in environmental fluctuation. Namely, a mutant phenotype that increases the reproductive output relative to others when the environment is generally favorable is likely to suffer a decline in reproductive output relative to others during an unfavorable period. Such a mutant allele increases the variance of the offspring number over time while the mean may remain unchanged. For example, as postulated by [26], a mutation that makes a female bird lay 10 eggs in one batch instead of spreading them over time in a randomly and rapidly fluctuating environment will increase the variance of her fitness while the mean remains the same. Setting the fitness of a mutant to $1 + S_t$ with $E[S_t] = 0$, thus an AMF mutant, models such a trade-off. Classical theory predicts that such a mutant is eliminated from a population as its geometric mean fitness is less than 1.

The previous [15] and this study, however, found that an AMF mutant at a low frequency can be positively selected (with both soft and hard selection) if a positive correlation between demographic and fitness fluctuations exists in the presence of a refuge whose carrying capacity oscillates in a narrower range and/or out of phase relative to the field. A heuristic argument for this NSE on the fitness-amplifying mutant allele may be given. In the TP model, the mean number of offspring per parent, *i.e.,* the absolute fitness, of the rare mutant in the field is given by $W_{2F} = \frac{K_F^{(t)}}{\bar{N}_F^{(t)}}(1 + S_t)$ (Eq 1), where $\frac{K_F^{(t)}}{\bar{N}_F^{(t)}}$ is the absolute fitness given under demographic fluctuation alone. With a positive correlation between $K_F^{(t)}$ and $S_t$, the fluctuation of $W_{2F}$ becomes greater than that of $1 + S_t$ and, more importantly, its arithmetic mean can now exceed 1. Still, the geometric mean of $W_{2F}$ is not greater than 1 due to its fluctuation over time. However, the presence of a refuge can reduce the variance of the mutant's fitness averaged across the whole population such that the geometric mean of this average fitness becomes greater than 1. Here, to generate this dampening effect, the refuge should not undergo the same fluctuation in size with the field; the absolute fitness in the refuge $W_{2R} = \frac{K_R^{(t)}}{\bar{N}_R^{(t)}}(1 + S_t^*)$ in Eq (1) should fluctuate differently than $W_{2F}$. With parameter values used in of the TP model, $K_R^{(t)}$ was given to covary incompletely with $K_F^{(t)}$. In the LSA model, the fluctuation of the refuge (the adult stage) is uncorrelated to the fluctuation of the field (the subadult stage). In [15], the seed bank fluctuates in size similar to the subadult and adult stages in the LSA model. In summary, such refuges, or noncovarying storages, turn the heightened arithmetic mean of the mutant allele's absolute fitness, raised by the demographic fluctuation of the field, into an elevation of the geometric mean in the total population.

Not all refuges in the previously proposed mechanisms of the storage effect are noncovarying. For example, it was suggested that if a gene is expressed in one sex in a dioecious species the other sex could be a refuge [9,15]. However, as equal numbers of male and female gametes are used in reproduction, the subpopulation sizes of reproductive males and females are effectively equal while the sex ratio of the adults may fluctuate. Therefore, as the size of the field relative to the refuge does not fluctuate at all, the NSE should not arise for the genes of the sex-limited expressions.

One may also consider GMF mutations; the fluctuating fitness of a mutant can be given by $exp[S_t]$, for example, making the mutant quasi-neutral as its geometric mean fitness is identical to that of the wild-type allele even without demographic fluctuation. Analyzing the model of cyclic environmental oscillation, the previous study [15] showed that the parameter range of positive selection for a rare GMF allele is wider than that for an AMF allele. Similarly, GMF alleles should be subject to the NSE under broader conditions. However, a GMF allele may be considered simply a beneficial allele as the arithmetic mean of offspring number is greater than the wild-type allele. Such a mutation is probably rare and therefore was not considered in this study. One should however note that GMF alleles might still be important for genetic variation in nature since they can be maintained in balanced polymorphism in the conventional models of the storage effect, for example, without demographic fluctuations and/or in the case of sex-limited gene expressions.

Analytic approximation for the one-locus TP model predicts that, under soft selection or strict density regulation that keeps the population size to the carrying capacity of the field at a given time regardless of its genetic composition, balancing selection arises if the demographic fluctuation (a major contributor to $Cov\left[S,\ log\left[\frac{K_F}{K_R}\right]\right]$) is not too weak or too strong compared to fitness fluctuation ($Var[S]$ in case of $S^* = 0$). Multi-locus simulations showed that balanced polymorphism can be simultaneously maintained at many loci, as expected since the demographic fluctuation (together with the presence of refuge) that turns the geometric mean fitness of an AMF allele from < 1 to > 1 should apply equally to all loci in the genome. Previously, Park and Kim [19] showed that, in the presence of a refuge but without fluctuation in population size, polymorphism can arise at multiple loci that contribute to the expression of a phenotype under fluctuating selection. However, the number of loci where balanced polymorphism emerged increased very slowly as the total number of loci in the model increased. In contrast, in the current simulation, balanced polymorphism was observed at all loci available for mutations to arise. However, the model containing only up to 10 loci was examined, due to the limitation of the simulation method used, and the 10-loci polymorphism at each locus was generally lower than the level observed in the one-locus simulation. This is probably because the segregation of alleles at multiple loci increases variance in the offspring number well above that of the Poisson distribution, thus increasing genetic drift that weakens the positive selection on rare alleles. The possibility that this mechanism can maintain the genome-wide polymorphism over 1,000 loci, currently observed in several natural populations [18,20], should be addressed by a simulation on a large scale in terms of both the number of loci and the census population size.

If the mutant ($A_2$) alleles that were positively selected from low to high frequencies are not maintained as a stable, oscillatory polymorphism as described above, they are fixed in the population. Such events, the fixations of mutations that amplify fitness fluctuations, are another outcome of the NSE with eco-evolutionary significance. In the TP model, the condition for the fixation is less restrictive compared to that for balanced polymorphism, because it requires relatively weaker selection (smaller variance in relative fitness) for a given demographic fluctuation to satisfy, assuming $S^* = 0$, $\Phi > 1 + 1/(2\rho)$ with soft selection and $\Phi > 1/(2\rho)$ with hard selection. Similar ranges of conditions (for $\Phi'$) seem to apply for the LSA model (Table C in S1 Table). Because fixation rather than balanced polymorphism occurs in a wider parameter space, one may argue that the overall effect of NSE (i.e., the modification of the storage effect due to population size fluctuation) is to make balancing selection less likely to occur. However, given the possible patterns of mutant alleles' fitness fluctuations in nature beyond those considered so far, the NSE may be a force of increasing, rather than decreasing, the number of loci in which oscillatory polymorphism occurs.

First, the NSE causes all GMF mutants to be either lost or fixed in the population, thus no balanced polymorphism [15], but allows AMF mutants to be maintained as balanced polymorphism under soft selection. Given that the net effects of

GMF mutants are to increase the arithmetic mean fitness above 1, and that random phenotypic changes are more likely to decrease rather than increase fitness [27], one may expect more AMF mutations than GMF mutations to occur in the genome. Therefore, the number of loci under balancing selection may increase due to the NSE. This also suggests that which pattern of fitness fluctuation allows the mutant allele to be "picked up" and maintained in polymorphism depends on the amplitude of demographic fluctuation. In the heuristic argument above, the absolute fitness of the $A_2$ allele in the field is given by $W_{2F} = D_F w_2$, where $D_F$ is its absolute fitness determined by a change in the population size of the field alone and $w_2$ is the relative fitness of $A_2$ in the field (e.g., in the TP model with AMF mutants, $D_F = \frac{K_F^{(t)}}{N_F^{(t)}}$ and $w_2 = 1 + S_t$). The arithmetic mean of $W_{2F}$ over time, $\overline{W_{2F}}$, which increases as positive correlation between $D_F$ and $w_t$ increases, determines the fate of the $A_2$ allele. When a wide fluctuation in $D_F$ occurs and the correlation between $D_F$ and $w_t$ is strong, $\overline{W_{2F}}$ may become very large, leading to the fixation of $A_2$. To reduce $\overline{W_{2F}}$ to a moderate value that is required for polymorphism, either the correlation between $D_F$ and $w_t$ (thus $\Phi$) or the arithmetic mean of $w_t$ needs to be reduced. The latter possibility implies that, for a strong demographic fluctuation under which even an AMF mutant is driven to fixation, a mutant with a new pattern of fitness fluctuation not considered in this study so far, possibly with even lower arithmetic mean than AMF mutants (thus < 1), might be picked up by the NSE into balanced polymorphism. If true, the NSE may greatly increase the level of genetic variation in species by generating polymorphism from an abundant class of phenotype-affecting mutations.

Demographic fluctuation that generates the NSE should apply equally to all loci in the genome. Therefore, fixations occur at all available loci on which fitness-amplifying mutations (*i.e.*, the $A_2$ allele) arise, as observed in the multi-locus simulation. With soft selection, the pattern of demographic fluctuation that generates the NSE does not change over time regardless of any evolutionary change at any locus. However, with hard selection, fluctuation in the relative size of the field generating the NSE at a given locus intensifies as more fluctuation-amplifying mutations reach fixation at other loci. Since the carrying capacity of the field in the TP model is effectively $(1 + S_t)^l K_F^{(t)}$, where $l$ is the number of loci where $A_2$ reached fixation, and wider fluctuation generates stronger positive selection on the rare $A_2$ (larger $E[\Delta_2]$ in S1 Appendix), fixation at one locus facilitates the subsequent fixations at other loci. This positive feedback between demographic fluctuation and positive selection drastically increases the amplitude of fluctuation in the total population size (Fig 4). How much the amplitude will increase should depend on the number of loci at which mutations subject to hard selection arise. There should be multiple independent ways in which changes in mutants' relative fitness can be translated into changes in the mean absolute fitness of the entire population. For various life-history traits, one may imagine phenotypes (smaller eggs, smaller adult body, earlier onset of first reproduction, etc.) that increase the absolute number of surviving offspring during a favorable, resource-rich period but decrease the number in a bad period while not affecting the amount of resources that the carriers of the alternative, ancestral alleles acquire. Then, many loci may be available for hard selection, the combined effect of which may lead to a very large fluctuation in population size.

The size of a population and its fluctuation should be governed not only by extrinsic factors (the environmental variables such as temperature and parasite prevalence) but also by intrinsic factors (species-specific responses to those extrinsic factors). The intrinsic factors such as various life-history traits are the product of the species' evolutionary history. For example, phenotypic plasticity responding to environmental cues can reduce the amplitude of the population size fluctuation, thus acting as a mechanism of population regulation [28,29], which according to the classical theory is adaptive as it enhances long-term reproductive success by increasing the geometric mean fitness [30–32]. If life-history traits evolved in the direction toward suppressing random or cyclic fluctuations in the population size [32], fluctuations observed in nature should be interpreted as the effect of variable environments that the evolved mechanisms of population regulation were unable to contain. However, this study suggests that evolution can proceed in the opposite direction. A large random fluctuation in population size can be the product of adaptive evolution driven by positive feedback between demographic and selective fluctuations. A recent study also suggested that a species may evolve to undergo a larger fluctuation in its population size. Liu *et al.* [33] found that a reproductive strategy increasing the temporal variance of offspring number can be advantageous in populations reproducing in overlapping generations under various patterns of environmental

fluctuations. When the degrees of population size fluctuation are different across species, the eco-evolutionary dynamics proposed in this study might be particularly important for those exhibiting large fluctuations, for example, mast seeding or insect outbreaks [34–36]. Interestingly, annual desert plants experiencing wider fitness fluctuations after germination have greater proportions of their population remaining in the seed bank [37]. This observation is in agreement with the analytic results in this study because a larger seed bank corresponds to a larger value of ρ, which makes it easier for the fixation of the fluctuation-amplifying alleles (i.e., lowering the threshold of Φ for positive selection) by the NSE.

## Acknowledgment

I thank Kristan Schneider, Claus Vogl, Jason Wolf, Sally Otto, and anonymous reviewers who read the previous and earlier versions of the manuscript and provided comments that greatly improved the scope and depth of this study.

## Supporting information

**S1 Appendix. Derivation of conditions for balancing and directional selection on A2 in the TP model.** (PDF)

**S2 Appendix. Methods of stochastic simulation.** (PDF)

**S1 Table. A. One-locus simulation with variable migration rates in the TP model; B. Mean time to the fixation ($q > 0.9$) of $A_2$ in the multi-locus simulation of the TP model; C. Long-term polymorphism in multi-locus simulations of the LSA model.** (PDF)

**S1 Fig. A. Exemplary trajectories of allele ($A_2$) frequencies in simulations; B. Examples of simulation runs in which the mutant allele ($A_2$) reaches fixation sequentially at all 8 loci used in the TP model; C. Exemplary allele frequency changes in the simulation of the LSA model; D. Mean fixation time ($\overline{\overline{T}}$) with varying number of loci (A; $L = 1$, 2, and 4) and the coefficient of variation in fixation times with $L = 4$ in the multi-locus simulation of the LSA model.** (PDF)

## Author contributions

**Conceptualization:** Yuseob Kim.

**Formal analysis:** Yuseob Kim.

**Investigation:** Yuseob Kim.

**Writing – original draft:** Yuseob Kim.

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
