## [Decision Letter · Decision Letter 0]

PONE-D-24-54939Noncovarying storage effect: balancing and positive directional selection on mutant alleles that amplify random fitness and demographic fluctuationsPLOS ONE

Dear Dr. Kim,

Thank you for submitting your manuscript to PLOS ONE. After careful consideration, we feel that it has merit but does not fully meet PLOS ONE’s publication criteria as it currently stands. Therefore, we invite you to submit a revised version of the manuscript that addresses the points raised during the review process.

While both reviewers agree that there is some merit in th earticle reviewer#1 suggests a change on the pitch of the article, while reviewer #2 founds the current version unsuitable for publication. As I found the reviewers comments constructive, I would like to ask you to revise your manuscript. However, this wil be a major revision, with substantial changes reqired. Particularly the language needs to be adjusted for better readability and the mathematics must be sound. As reviewer #2 suggests rejection, the manuscript must be a substantially improvemet, with a clear presentation and rigorous mathematics. Reviewer #2 mentiones that some frustration is accompaining the manuscript.However, I believe that both reviewers make constructive comments that will guide a substantial revision, which should be regarded an opportunity rather than a cause for more frustration.

We look forward to receiving your revised manuscript.

Kind regards,

Kristan Alexander Schneider, Ph.D.

Academic Editor

PLOS ONE

Journal Requirements:

3. Thank you for stating the following financial disclosure: This study was supported by the National Research Foundation (NRF) grant 2020R1A2C1009261 funded by the Korean government. 

4. Thank you for stating the following in the Acknowledgments Section of your manuscript: This study was supported by the National Research Foundation (NRF) grant 2020R1A2C1009261 funded by the Korean government.

Please remove any funding-related text from the manuscript and let us know how you would like to update your Funding Statement. Currently, your Funding Statement reads as follows: This study was supported by the National Research Foundation (NRF) grant 2020R1A2C1009261 funded by the Korean government.

Reviewers' comments:

Reviewer's Responses to Questions

**Comments to the Author**

1. Is the manuscript technically sound, and do the data support the conclusions?

Reviewer #1: No

Reviewer #2: Partly

2. Has the statistical analysis been performed appropriately and rigorously? 

Reviewer #1: Yes

Reviewer #2: N/A

3. Have the authors made all data underlying the findings in their manuscript fully available?

Reviewer #1: Yes

Reviewer #2: Yes

4. Is the manuscript presented in an intelligible fashion and written in standard English?

Reviewer #1: Yes

Reviewer #2: No

5. Review Comments to the Author

Reviewer #1: The author studies the maintenance of variation under fluctuating selection. Using two different population models, two-patch model and larval-subadult-adult model with hard and soft selection. The work shown in this manuscript is a generalization of previous results of the author. While the analyses are complete and convincing, the reviewer is concerned about the pitch of the manuscript.

My understanding is that variation is maintained only under a set of very restricted conditions, hence the focus should be on the difficulty to maintain variation under fluctuating selection, rather than portraying it as a realistic scenario. As written, the reviewer feels that the study may be cited for evidence that variation is maintained under fluctuating selection, although most results show the opposite.

It may also help to make more effort to discuss how biologically reasonable the permissive parameter space is. In other words, how likely is it that these conditions are met in natural populations?

Reviewer #2: The present article spans the fields of population genetics and ecology and, thus, should be considered interdisciplinary. The file includes comments of previous reviewers and the reply of the author. It seems that the reviewers as well as the author were quite frustrated. I do share the sentiments of the author: Like the author, I disagree "with the assessment ... that the current manuscript is not a sufficient advancement over Kim (2023)", for the reasons given by the author. I also do not think the article is too mathematical or technical and should rather be submitted to eg the journal "Theoretical Population Biology". However, I think that the present version is far from publishable. In this respect, I agree with some of the reviewers and share their frustration.

As this is an interdisciplinary study, I should maybe specify my background: I have published in population genetics, both articles that were data-heavy as well as theory-heavy. I should have no problems following the math in the present article. I have worked and published in various fields as a statistical consultant and co-written articles in those fields. I am therefore very aware of the difficulties working in interdisciplinary contexts and, especially, how much care and thinking must be invested in appropriately identifying the relevant readers, in choosing the right language, and in finding the correct terms and definitions that are acceptable in all relevant fields. It is also important to be very clear and precise, because otherwise readers of one or the other field may be lost completely. An example for an article that achieves this feat admirably is that of Bruce Wallace (1975), cited in the present article.

Unfortunately, the present article is very confusing. Like one of the earlier reviewers, I found the abstract already problematic; eg: "Recent study found that, under pre-existing oscillation in population size, the mutant is positively selected to fixation if its fitness change correlates with the rate of population growth, which further amplifies population size oscillation." This sentence should likely be split into three or more sentences. For a population geneticist the fitness of a mutant compared to the wild-type (allele, since haploid individuals are assumed throughout) would make sense. What then does the "fitness change" of a mutant allele mean? And why should that correlate with the population size? An oscillating population size should not grow (on average); so why "the rate of population growth"? Hence, already the abstract confused me. The introduction did not help either, so I hoped for some clarity when the mathematical models were described. But here I was also left confused:

Fig 1. should provide an illustration for the model. I found the graphs reasonable, but the legend problematic, eg: "Reproduction, followed immediately by mutation and reproduction,..." Here the second "reproduction" makes no sense to me.

Before the passage starting at l202, populations sizes in the two patches, field $F$ and refuge $R$, were described and denoted with symbols $N_F^{(t)}$ and $N_R^{(t)}$. Obviously, these are both finite. But consider: "The model so far described changes in the number of individuals without genetic drift. When a finite population size is assumed, as in simulation below, each parent in both subpopulations is modeled to produce a Poisson number of offspring with the mean specified above, which approximates genetic drift in the Wright-Fisher model." As genetic drift is zero only if the (effective) population size is infinite, the first sentence makes no sense, given that just before the finite population sizes $N_F^{(t)}$ and $N_R^{(t)}$ were described and effective population sizes are usually much smaller than census population sizes. Furthermore, the Wright-Fisher (WF) model usually considers diploid individuals; without selection a binomial distribution with mean offspring number of 2 per individual is assumed implicitly. If a haploid version of the Wright-Fisher model is assumed (as would be appropriate for the assumptions of the author), the offspring distribution per individual is again binomial with mean one. For large population sizes, the binomial distribution could be approximated by a Poisson distribution. In both cases, the strength of drift is inversely proportional the (effective) population size. In any case, the statement that "a Poisson number of offspring" approximates "genetic drift in the Wright-Fisher model" is at least confusing, if not simply wrong.

Close to the "heart" of the model is the stochastic variation of the key parameters $S_t$, $S_t^*$, $U_t$ and $V_t$. We are informed that the expectations of all these variables are 0, and that the variances of $S_t$ and $S_t^*$ are positive (in the paragraph starting at l158). But we are not informed about the distribution of these two variables. On the other hand, we are informed that $U_t$ and $V_t$ are drawn from a joint normal distribution (paragraph starting with l207) without information on their variance-covariance matrix. Immediately, below an inequality relation of the covariances is given: $\Cov(S_t,U_t-V_t) > \Cov(S_t^*,U_t-V_t) $. Should we deduce from this that all four variables are jointly normally distributed with zero means and some arbitrary Var-Cov matrix, as long as the inequality is fulfilled? Actually in formula 2, $\Cov(S,S*)$ appears as a term. So: maybe yes?!?

There is a further problem with $S_t$ and $S_t^*$: Should $S_t$ be less than 1, $1- S_t$ is negative and therefore the carrying capacity would also become negative (see the formulas in Appendix A), which is non-sensical, but if the two parameters are taken from a normal distribution this would be possible as the support of the normal distribution is from minus to plus infinity. The solutions would be the same as for the variables $U_t$ and $V_t$: assume a log-normal distribution for $S_t$ and $S_t^*$ and then to use $e^{S_t}$ instead of $1-S_t$ (the former approximates the latter anyway as long as $S-t$ is small). As the problem of negative carrying capacities may have messed up the simulations the author might have assumed that anyway. If the mathematics would have been described with the necessary rigor, including the description of the support of the random variables, this would have become obvious right away.

The analytical results presented in Appendix A are deterministic deterministic in the allele frequencies. This fact should be pointed out in the main text. Note that this assumption leads to the paradoxical situation that a fluctuating population size is assumed to be infinitely large (such that the drift can be assumed to be zero). This problem (that I already stumbled on above) should be discussed.---In l333ff, the author actually shows that the simulations can reproduce the key predictions of the deterministic analysis. Given the assumptions, this should break down if the population sizes are very small and drift correspondingly large. Indeed, this seems to be confirmed, as the results for small population sizes are indistinguishable from neutrality. Hence, the analytical and simulation results seem to agree to the expected degree, which is reassuring.

Generally, this impression of reassuring solidity hidden beneath a sloppy verbal description held for much of the results section; the discussion also seems generally OK.

In summary, the article is far from readable. At least this reader with a popgen, math, and interdisciplinary background could not follow key parts in the abstract, the introduction and the model sections. Nevertheless, the analytic and simulation results are interesting, agree sufficiently to appear solid and seem interesting enough for eventual publication. The discussion also contained some useful further information.

Some random confusion:

l193: "First, under soft selection, very strong density regulation occurs so that the final

size of the field is constrained to $ _ $." The strength of density regulation under soft selection depends on the absolute number of individuals before density regulation. If the latter number is below the maximal capacity, no individuals would need to die or be excluded from reproduction. As a surplus of offspring is a prerequisite for evolution, this case may be considered degenerate, but at least possible.

In the paragraph starting at l207: "The oscillation of carrying capacities is therefore symmetrical in the log scale, which is generally predicted in demographic models and observed in nature..." I guess that the symmetry is assumed (not predicted) in theoretical demographic models. And I consider it impossible to "observe" this in nature, although (most) data may be "consistent with" the assumption.

Why is it assumed that subadults are reproducing? I would define adults as individuals that have reached reproductive age, while subadults are below that. Maybe "young adults" and "old adults" would make more sense here. This model seems similar to that presented by Cenik and Wakeley (2010) PLoS One 5(9):e13019. (But those authors only consider the popgen part, not the ecological part or the feedback between the two as the present author. This shows again that the basic idea of the author is indeed interesting and worthwhile to follow up.)

6. PLOS authors have the option to publish the peer review history of their article (what does this mean? ). If published, this will include your full peer review and any attached files.

**Do you want your identity to be public for this peer review?** For information about this choice, including consent withdrawal, please see our Privacy Policy .

Reviewer #1: No

Reviewer #2: **Yes: ** Claus Vogl

---

## [Author Response · Author response to Decision Letter 1]

25 Feb 2025

Response to reviewer comments are found in the "POneResonsToReview.pdf" file that was submitted along with the revised mansucript.

---

## [Decision Letter · Decision Letter 1]

PONE-D-24-54939R1Noncovarying storage effect: balancing and positive directional selection on mutant alleles that amplify random fitness and demographic fluctuationsPLOS ONE

Dear Dr. Kim,

Thank you for submitting your manuscript to PLOS ONE. After careful consideration, we feel that it has merit but does not fully meet PLOS ONE’s publication criteria as it currently stands. Therefore, we invite you to submit a revised version of the manuscript that addresses the points raised during the review process.

One of the original reviewers is satisfied with the revision wheas the other (reviewer#1) raised concerns that his original comments were not adequately addressed ("My understanding is that variation is maintained only under a set of very restricted conditions, hence the focus should be on the difficulty to maintain variation under fluctuating selection, rather than portraying it as a realistic scenario."). The additional reviewer was very positive about the manuscript. I also went over it carefully myself. I want to point out at this point that PLOS ONE's policy is to publish research based on scientific rigor, rather than impact, novelty, or relevance. Having said that, I see the point of reviewer #1. My own impression is that the NSE is motivated by eleology, but the model analyzed is haploid. Hence my impression was that there is a bit of a stretch in promoting NSE as a common and relevant mechanism. Hence, I would like to invite you to submit a revised version of the manuscript, in which you focus more on the mechanism itself and intuitive explanations of why NSE serves as a mechanism of balancing selection rather than promoting its relevance in ecology. It would be good if you could give some specific examples on when NSE can arise for haploid organisms. I als think you could add some intuitive explanations of how the underlying mechanism works in some places. Since two reviewers were satisfied, please keep in mind to keep the revision "minimally invasive".

We look forward to receiving your revised manuscript.

Kind regards,

Kristan Alexander Schneider, Ph.D.

Academic Editor

PLOS ONE

Journal Requirements:

Reviewers' comments:

Reviewer's Responses to Questions

**Comments to the Author**

1. If the authors have adequately addressed your comments raised in a previous round of review and you feel that this manuscript is now acceptable for publication, you may indicate that here to bypass the “Comments to the Author” section, enter your conflict of interest statement in the “Confidential to Editor” section, and submit your "Accept" recommendation.

Reviewer #1: (No Response)

Reviewer #2: All comments have been addressed

Reviewer #3: All comments have been addressed

Reviewer #4: All comments have been addressed

2. Is the manuscript technically sound, and do the data support the conclusions?

Reviewer #1: Partly

Reviewer #2: Partly

Reviewer #3: Yes

Reviewer #4: Yes

3. Has the statistical analysis been performed appropriately and rigorously? 

Reviewer #1: N/A

Reviewer #2: N/A

Reviewer #3: Yes

Reviewer #4: Yes

4. Have the authors made all data underlying the findings in their manuscript fully available?

Reviewer #1: Yes

Reviewer #2: Yes

Reviewer #3: Yes

Reviewer #4: Yes

5. Is the manuscript presented in an intelligible fashion and written in standard English?

Reviewer #1: Yes

Reviewer #2: No

Reviewer #3: Yes

Reviewer #4: Yes

6. Review Comments to the Author

Reviewer #1: I was not convinced that the minor changes to the text reflect my different perception of the message. Hence, I think that the results as presented portray the wrong impression and the ms may be cited for the misleading message.

Reviewer #2: General:

The writing has been improved and I could generally understand the article. The improved clarity has uncovered only minor problems with the underlying science. The manuscript still needs editing to improve language and readability and, in places, mathematical rigor (non-exhaustive comments mainly in the attached text).

I have one general point to make: It seems to me that one side is ignored, which should be treated for the sake of clarity and completeness of the argument. I will start out with a term that is unclear: "incomplete correlation". A correlation is only rarely exactly plus or minus one (if this is meant with "complete"); is an "intermediate correlation" meant, maybe between plus or minus 0.2-0.7. While "complete correlation" may make sense (a correlation of exactly plus or minus one) and an "intermediate correlation" may be between plus or minus 0.2-0.7, "covary incompletely", i.e, an "incomplete covariance" makes less sense, as far as I can see.

This goes beyond picking words: As far as I understand the correlation (in population size between the field and refuge) must be *positive* and intermediate; a *negative* correlation would not allow for the major point of the article. I guess that instead of balancing selection (and thus balanced polymorphism), disruptive selection should result from a negative correlation. If this is true, this possibility needs to be mentioned and the parameter ranges, where it can occur, need to be explored. I guess that this case is of little biological relevance but, at least to me, would be required to make the argument complete.

Specifics:

Sometimes I had trouble with the wording (non-exhaustive comments in the text).

I am not sure if "fitness-amplifying" is the best term, but reconciled myself to it.

I am unclear in places, eg, l207, whether an absolute frequency (a count) or a proportion is meant.

l 174 I would say that selection does not require genetic variation, but evolution (the response to selection) does.

l204 I think there is a logical error here.

and see many other comments in the text...

Reviewer #3: I am the third reviewer who involved in the second round review.

I read through the manuscript and the response letter. I like the revised manuscript very much. I personally think that the author has addressed all concerns that were raised by the two reviewers. After the author made the revisions, the manuscript is much clearer than the first version.

In the revised manuscript, the author studied a very important question about how natural selection occurs in natural populations. Noncovarying storage effect is straightforward to describe the effect. I also like the theoretical analysis, one-locus simulation and multi-locus simulation. Therefore, I would suggest the editor to accept the manuscript in the current form.

Reviewer #4: I am the third reviewer who involved in the second round review.

I read through the manuscript and the response letter. I like the revised manuscript very much. I personally think that the author has addressed all concerns that were raised by the two reviewers. After the author made the revisions, the manuscript is much clearer than the first version.

In the revised manuscript, the author studied a very important question about how natural selection occurs in natural populations. Noncovarying storage effect is straightforward to describe the effect. I also like the theoretical analysis, one-locus simulation and multi-locus simulation. Therefore, I would suggest the editor to accept the manuscript in the current form.

7. PLOS authors have the option to publish the peer review history of their article (what does this mean? ). If published, this will include your full peer review and any attached files.

**Do you want your identity to be public for this peer review?** For information about this choice, including consent withdrawal, please see our Privacy Policy .

Reviewer #1: No

Reviewer #2: **Yes: ** Claus Vogl

Reviewer #3: No

Reviewer #4: No

---

## [Author Response · Author response to Decision Letter 2]

13 Jun 2025

Please find the "response to reviewers" file submitted together with the revised manuscript.

---

## [Editor Report · Decision Letter 2]

Noncovarying storage effect: balancing and positive directional selection on mutant alleles that amplify random fitness and demographic fluctuations

PONE-D-24-54939R2

Dear Dr. Kim,

We’re pleased to inform you that your manuscript has been judged scientifically suitable for publication and will be formally accepted for publication once it meets all outstanding technical requirements.

Kind regards,

Kristan Alexander Schneider, Ph.D.

Academic Editor

PLOS ONE
---

## [Editor Report · Acceptance letter]

PONE-D-24-54939R2

PLOS ONE

Dear Dr. Kim,

I'm pleased to inform you that your manuscript has been deemed suitable for publication in PLOS ONE. Congratulations! Your manuscript is now being handed over to our production team.

Kind regards,

on behalf of

Professor Kristan Alexander Schneider

Academic Editor

PLOS ONE